# SN 2017fzw: A Fast-Expanding Type Ia Supernova with Transitional Features

**Jiayu Huang** [1,2], **Yangyang Li** [1,2,3], **Xiangyun Zeng** [1,2,*], **Sheng Zheng** [1,2,*], **Sarah A. Bird** [1,2], **Jujia Zhang** [4,5,6],
**Ali Esamdin** [7], **Abdusamatjan Iskandar** [7,8], **K. Azaleee Bostroem** [9,10], **Shuguang Zeng** [1,2], **Yanshan Xiao** [1,2],
**Yao Huang** [1,2], **D. Andrew Howell** [9,10], **Curtis McCully** [9,10], **Wenxiong Li** [11], **Tianmeng Zhang** [12], **Lifan Wang** [13]
**and Lei Hu** [14]

1    Center for Astronomy and Space Sciences, China Three Gorges University, Yichang 443000, China;
     huangjy1998@126.com (J.H.); liyangyang11223@163.com (Y.L.); sarahbird@ctgu.edu.cn (S.A.B.);
     zengshuguang19@163.com (S.Z.); xiaoyanshan@ctgu.edu.cn (Y.X.); huangyao@ctgu.edu.cn (Y.H.)
2    College of Science, China Three Gorges University, Yichang 443000, China
3    Mathematics and Science College, Shanghai Normal University, Shanghai 200233, China
4    Yunnan Astronomical Observatories, Chinese Academy of Sciences, Kunming 650216, China; jujia@ynao.ac.cn
5    Key Laboratory for the Structure and Evolution of Celestial Objects, Chinese Academy of Sciences,
     Kunming 650216, China
6    Center for Astronomical Mega-Science, Chinese Academy of Sciences, 20A Datun Road, Chaoyang District,
     Beijing 100012, China
7    Xinjiang Astronomical Observatory, Chinese Academy of Sciences, Urumqi 830011, China;
     aliyi@xao.ac.cn (A.E.); abudu@xao.ac.cn (A.I.)
8    School of Astronomy and Space Science, University of Chinese Academy of Sciences, Beijing 100049, China
9    Department of Physics, University of California, Santa Barbara, CA 93106-9530, USA;
     abostroem@gmail.com (K.A.B.); dahowell@gmail.com (D.A.H.); cmccully@lco.global (C.M.)
10   Las Cumbres Observatory, 6740 Cortona Drive Suite 102, Goleta, CA 93117-5575, USA
11   The School of Physics and Astronomy, Tel Aviv University, Tel Aviv 69978, Israel; li-wx15@tsinghua.org.cn
12   Key Laboratory of Optical Astronomy, National Astronomical Observatories, Chinese Academy of Sciences,
     Beijing 100012, China; zhangtm@nao.cas.cn
13   George P. and Cynthia Woods Mitchell Institute for Fundamental Physics & Astronomy, Department of
     Physics and Astronomy, Texas A&M University, 4242 TAMU, College Station, TX 77843, USA; lifan@tamu.edu
14   Purple Mountain Observatory, Nanjing 210023, China; hulei@pmo.ac.cn
*    Correspondence: xyzeng2018@ctgu.edu.cn (X.Z.); zsh@ctgu.edu.cn (S.Z.)

**Abstract:** In this study, we analyzed the optical observations of a subluminous Type Ia supernova (SN Ia) 2017fzw, which exhibited high photospheric velocity (HV) at $B$-band maximum light. The absolute $B$-band peak magnitude was determined to be $M^B_{max} = -18.65 \pm 0.13$ mag, similar to 91bg-like SNe Ia. An estimation of the rate of decline for the $B$-band light curve was determined to be $\Delta m_{15}(B) = 1.60 \pm 0.06$ mag. The spectra of SN 2017fzw were similar to those of 91bg-like SNe Ia, with prominent Ti II and Si II $\lambda5972$ features at early phases, gradually transitioning to spectra resembling normal (mainly HV subclass) SNe Ia at later phases, with a stronger Ca II NIR feature. Notably, throughout all phases of observation, SN 2017fzw displayed spectral evolution characteristics that were comparable to those of HV SNe Ia, and at peak brightness, the Si II $\lambda6355$ velocity was determined to be 13,800 $\pm$415 km s$^{-1}$ and a more pronounced Ca II NIR feature was also detected. Based on these findings, we classify SN 2017fzw as a transitional object with properties of both normal and 91bg-like SNe Ia, providing support for the hypothesis of a continuous distribution of supernovae between these two groups.

**Keywords:** supernovae: general; supernovae: individual: SN 2017fzw; transitional supernovae

## 1. Introduction

Type Ia supernovae (SNe Ia) are highly valued cosmological distance indicators [1] that have played a pivotal role in discovering the accelerating expansion of the universe [2] due

to their uniform light curves and high luminosities. Explosive thermonuclear runaways are commonly thought to be the cause of these supernovae, which are believed to originate from white dwarfs (WDs) composed of carbon and oxygen [3]. Typically, their close-binary companion star is considered a non-degenerate object, known as the single-degenerate scenario (SD) [4,5], or another WD, known as the double-degenerate scenario (DD) [6]. Observations support both the SD and DD scenarios, which are both used to explain the observed diversity among SNe Ia [7,8].

The majority of SNe Ia ($\sim$70%) are classified as Branch normal groups, which exhibit relatively uniform photometric and spectroscopic evolution [9,10]. The remaining SNe Ia are grouped into various subclasses based on differences in their photometric and spectroscopic evolution compared to normal SNe Ia. The subclass of SN 1991T-like, which is excessively bright, can be distinguished by the presence of wide light curves, relatively feeble Si II/S II absorption features, and conspicuous Fe II/Fe III absorption features in the vicinity of the point of maximum brightness [11–13]. On the other hand, the subluminous SN 1991bg-like subgroup, such as SN 1999 by [14], display a rapid decline rate in their light curves and strong absorption features of intermediate-mass elements (IMEs) [15,16]. In most cases, the rapid decliner subclass of SNe Ia is accompanied by low peak luminosity and accounts for approximately 15–20% of all observed SNe Ia [17–19]. Moreover, the rapid decliner class has a single *I*-band maximum delayed a few days with regard to the *B*-band maximum. For the 91bg-like group, the above behavior is mainly due to their lower mass of $^{56}$Ni [20,21]. Transitional Type Ia supernovae exhibit photometric and spectroscopic characteristics that lie between those of normal Type Ia supernovae and 91bg-like Type Ia supernovae [22]. These transitional SNe Ia appear with a frequency as high as 91bg-like SNe Ia, such as SNe 1986G [23], 2011iv [24], and 2012ij [25].

Gaining insights into the characteristics of transitional Type Ia supernovae, which lie intermediate between those of normal Type Ia supernovae and 91bg-like Type Ia supernovae, is of paramount importance in comprehending their underlying origin. The identification of transitional SNe with a continuous distribution of observed properties supports the theory that normal and 91bg-like SNe share a common origin [26]. Transitional SNe Ia are characterized by a large light curve shape parameter ($\Delta m_{15}(B)$, the decline rate in *B*-band light curve over 15 days after the peak [27,28]), and low luminosity, which are similar to 91bg-like SNe [24,26]. However, their near-infrared (NIR) light curves resemble those of normal SNe Ia that exhibit two maxima [24,29]. Spectroscopically, transitional SNe Ia show similarities to both normal and 91bg-like SNe Ia, which demonstrates their intermediate properties. Previous studies have reported that the range of the Phillips parameter for transitional groups is typically 1.5–1.8 mag [19,30,31]. However, it is important to note that some SNe Ia beyond this range are also classified as transitional SNe, while some SNe Ia within this range do not exhibit any clear intermediate properties [20,32].

There are previous studies for the observed properties of SN 2017fzw, Sand et al. (2019) found that SN 2017fzw was like the subluminous SN 2018fhw, which was not detected any H$\alpha$ emission at late times, indicating that it may have a non-degenerate companion [33]. Then, Tucker et al. (2020) found that SN 2017fzw did not exhibit evidence of stripped companion emission [34]. Additionally, Graham et al. (2022) demonstrated the light curve parameters of SN 2017fzw measured from Las Cumbres Observatory (LCO) [35] and presented a deep analysis of SN 2017fzw with its observed spectra at early and nebular phases [36]. They found that SN 2017fzw seems to be a transitional object between normal and subluminous 91bg-like SNe Ia.

In this paper, we report on photometric and spectroscopic observations of SN 2017fzw, covering the new and previously published data. According to the new photometric datafrom *Neil Gehrels Swift Observatory* (*Swift*) [37] on SN 2017fzw, a full analysis of photometric and spectroscopic observations for this SN, we found that SN 2017fzw is a transitional SN Ia with high velocity (HV). In Section 2, we provide a detailed description of the observations and data reduction process. In Section 3, we present the optical light curves, color curves, estimation of reddening, and quasi-bolometric light curves of SN 2017fzw.

In Section 4, we predict its mean spectral sequences and compare the spectral evolution with other SNe Ia. We discuss our findings in Section 5 and present our conclusions in Section 6.

## 2. Observations and Data Reduction

SN 2017fzw was first detected on 9 August 2017, at 0.41-m PROMPT-5 telescope, located at Cerro Tololo Inter-American Observatory (CTIO) during the Distance Less Than 40 Mpc survey (DLT40) conducted by Tartaglia et al. (2018) [38]. Its coordinates were measured to be $\alpha = 06^h21^m34^s.820$ and $\delta = -27°12'53''.57$ (J2000), and its initial clear-filtered magnitude (AB) was recorded as approximately 17.17 mag [39,40]. Hosseinzadeh et al. (2017) [41] conducted a subsequent spectroscopic study, which identified SN 2017fzw as a 91bg-like SN Ia and determined its host galaxy to be NGC 2217 at a redshift of $z = 0.0054$. This corresponds to a distance modulus ($\mu$) of $\mu = 31.66 \pm 0.15$ mag, or a distance of 21.45 Mpc (taking into account only the Virgo Infall) with an assumed Hubble constant of $67.8 \, \text{km s}^{-1} \, \text{Mpc}^{-1}$ [42,43]. SN 2017fzw is located approximately 100 away from the center of its host galaxy. Figure 1 displays the finder chart of SN 2017fzw.

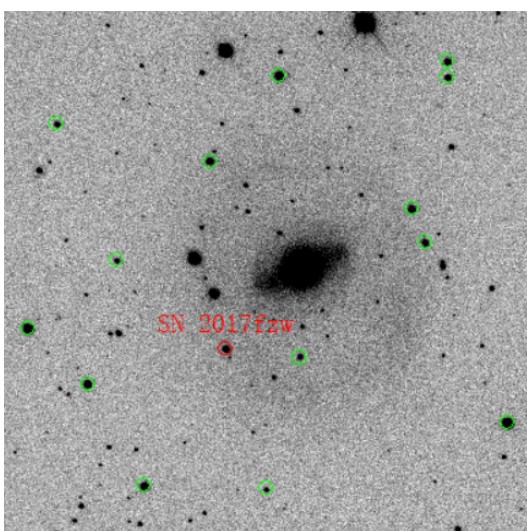

**Figure 1.** SN 2017fzw (red circle) and a local sequence of standard stars (green circles) in the field of NGC 2217 (*i*-band image, obtained by LCO on 7 September 2017 UT).

The optical photometry ($BVgri$) of SN 2017fzw was mainly collected by the 1 m telescopes ofLCO [35,44] network for the Global Surpernova Project [45]. The data of LCO were reduced using `lcogtsnpipe` [46] and a `PyRAF`-based pipeline. The instrumental magnitudes of LCO are calibrated relative to Landolt (1992) [47] ($BV$) and Smith et al. (2002) [48] ($gri$) standard stars observed over multiple photometric nights, which are detailedly introduced by Krisciunas et al. (2017) [49] and Phillips et al. (2019) [50]. Ultraviolet (UV) and optical observations of this SN were also observed withthe *Neil Gehrels Swift Observatory* (*Swift*) [37] in six bands, including $UVW1$, $UVW2$, $UVM2$, $U$, $B$, and $V$ filters [51]. The $UVM2$-band data could be ignored due to only four observations. According to the zero points of Breeveld et al. (2011) [52] in the Vega magnitudes, we obtained the *Swift* UV/optical light curves using the data-reduction pipeline of the *Swift* UV/optical Supernova Archive (SOUSA) [53]. We measured the source counts using a 3″ aperture and corrected by an average point-spread function. Thus, the final flux-calibrated LCO and *Swift* light curves of SN 2017fzw are listed in Table A1, and shown in Figure 2.

Following the discovery, optical observations were carried out using FLOYDS spectrographs that were installed on the 2 m Faulkes Telescope North and South of the LCO [35,54], which are from the Global Supernova Project [45]. Between −9 and +168 days, a total of 11 low-resolution optical spectra were acquired for this particular supernova. The spectral flux of SN 2017fzw are calibrated by standard stars observed with a comparable air mass as

the SN on the same night. The LCO extinction curves and telluric correction are utilized to correct for the effects of atmospheric extinction and telluric absorption lines in the spectral data of SN 2017fzw.

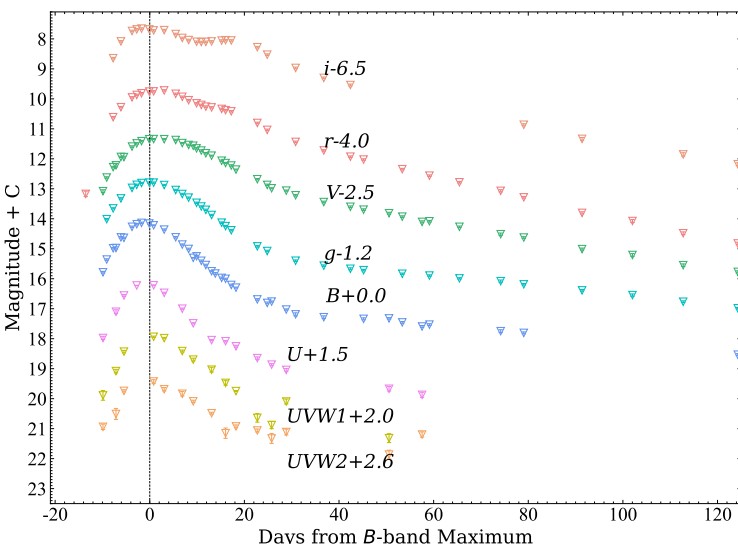

**Figure 2.** UV and optical (*UBVgri*) light curves of SN 2017fzw obtained by *Swift* and LCO. The vertical dashed line indicates its *B*-band maximum, and the light curves have been shifted vertically for clarity.

## 3. Photometric Properties

### 3.1. Optical Light Curves and Time of First Light

Figure 2 shows the UV/optical light curves of SN 2017fzw. These light curves have a nearly daily cadence from about $-10$ to $+125$ days relative to *B*-band maximum light. Like normal SNe Ia, the $r/i$-band light curves of SN 2017fzw clearly show a shoulder/secondary maximum, and its light curve peak reached slightly earlier in $i$-band relative to *B*-band. These multi-band light curves were fit by SuperNovae in the object oriented Python code SNooPy2 [55,56] to determine the time of *B*-band maximum light and other important light curve parameters, as shown in the left panel of Figure 3. Our investigation revealed that the *B*-band maximum light ($T_{max}(B)$) of SN 2017fzw occurred on MJD $57,987.90 \pm 0.35$, corresponding to a peak magnitudeof $B_{max} = 13.20 \pm 0.07$ mag and a decline rate $\Delta m_{15}(B) = 1.60 \pm 0.06$ mag estimated in the rest frame of this SN.

The time of first light (FLT) was estimated as MJD $57,972.00 \pm 0.24$ by fitting the early multi-band light curves of SN 2017fzw during the $\sim$7 days before the *B*-band maximum with the ideal expanding fireball model [57], as shown in the right panel of Figure 3. Thus, its rise time is given as $15.9 \pm 0.4$ days, which is comparable to the average value of normal SNe Ia (i.e., 16.0 days [58]) and also close to the upper limit value of 91bg-like SNe Ia ($\sim$13–15 days [20]). The basic photometric parameters of this SN are listed in Table 1, which are consistent with results of Graham et al. (2022) [36] within the uncertainties. For the following discussions, the phases are all given in reference to the *B*-band maximum of SN 2017fzw.

**Table 1.** The photometric and spectroscopic parameters of SN 2017fzw.

| Parameter | Value (This Paper) | Value (Graham et al. (2020) [36]) |
|---|---|---|
| $B_{max}$ | $13.20 \pm 0.07$ mag | $13.25 \pm 0.16$ mag |
| $M_{max}^B$ | $-18.65 \pm 0.13$ mag | $-18.81 \pm 0.18$ mag |
| $\Delta m_{15}(B)$ | $1.60 \pm 0.06$ mag | $1.60 \pm 0.02$ mag |
| $S_{BV}$ | $0.63 \pm 0.04$ | - |
| $T_{max}(B)$ (days) | MJD $57{,}987.90 \pm 0.35$ | $57{,}987.90$ |
| FLT (days) | MJD $57{,}972.00 \pm 0.24$ | - |
| Rise Time | $15.9 \pm 0.4$ days | - |
| $\mu$ | $31.71 \pm 0.15$ mag | $32.06 \pm 0.03$ mag |
| Redshift [41] | $0.0054$ | $0.0054$ |
| $v$ | $13{,}800 \pm 415$ km s$^{-1}$ | - |
| $M_{Ni}$ | $0.18 \pm 0.03$ M$_\odot$ | - |

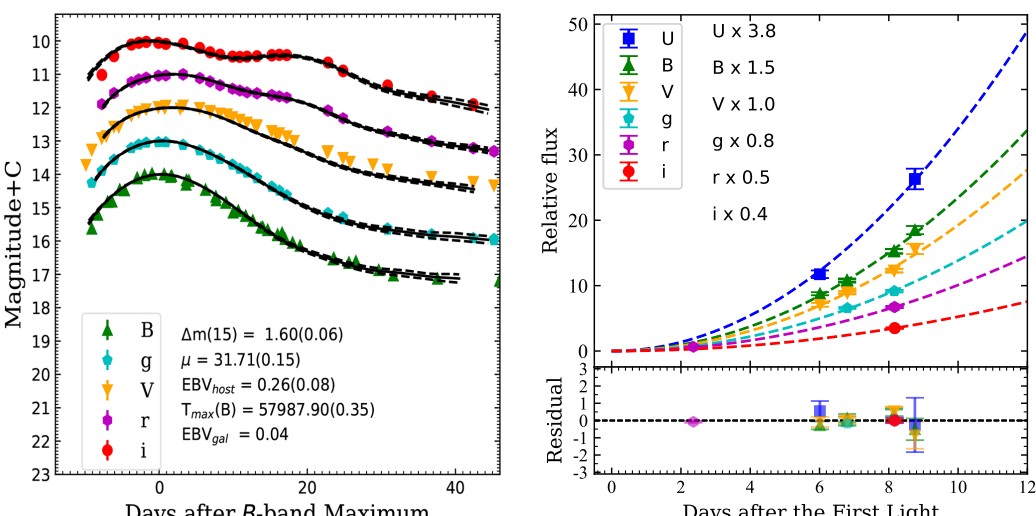

**Figure 3.** Left panel: best-fit light curve model (solid black lines) from `SNooPy2` [55,56] for SN 2017fzw. The dashed black lines indicate the $1-\sigma$ uncertainty (in many cases smaller than the line width) with respect to the best-fit light curve templates, and the light curves have been shifted vertically for clarity. Right panel: ideal fireball model [57] fits (dashed lines) to the multi-band early light curves of SN 2017fzw (markers with error bars) during the $\sim7$ days before the $B$-band maximum. The bottom panel displays the residual of the best-fit curves, and the horizontal black dashed line represents zero residual.

As Figure 4 shows, in this study we conducted a comparative analysis of the *BV*-band luminosity profiles of SN 2017fzw against those of a selection of normal and subluminous type Ia supernovae, including SNe 1986G [23,28,59], 1999by [14,60], 2004eo [61], 2011fe [62], 2012ij [25], and 2017fgc [63]. One can see that the light curves of SN 2017fzw, like other transitional SNe Ia, are all similar to those of SN 1999by in terms of their morphology. In particular, the light curve of SN 2017fzw in *V*-band shows the strongest resemblance to that of transitional SN 1986G having similar $\Delta m_{15}(B)$.

### 3.2. Reddening and Color Curves

According to the NASA/IPAC Extragalactic Database (NED) [43], the Galactic extinction toward SN 2017fzw is $A_B^{Gal} = 0.16$ [64]. Since the Na I D absorption feature was non-detectable in this SN spectra, the reddening of the host galaxy was assumed to be negligible. Fitting the multi-band light curves of SN 2017fzw with `SNooPy2` estimates its $\mu$ as $31.71 \pm 0.15$ mag (see the left panel of Figure 3). Taking $\mu$ of its host galaxy as $31.66 \pm 0.15$ mag into account, we adopt their average value of $\mu = 31.69 \pm 0.11$ mag and an extinction law $R_V = 3.1$ [65] in the following analysis. Then, the $B$-band absolute peak

magnitude of SN 2017fzw was deduced as $M_{max}^B = -18.65 \pm 0.13$ mag, which is located the absolute luminous end of the 91bg-like SNe Ia (about $-16.5$ to $-17.7$ mag [66]).

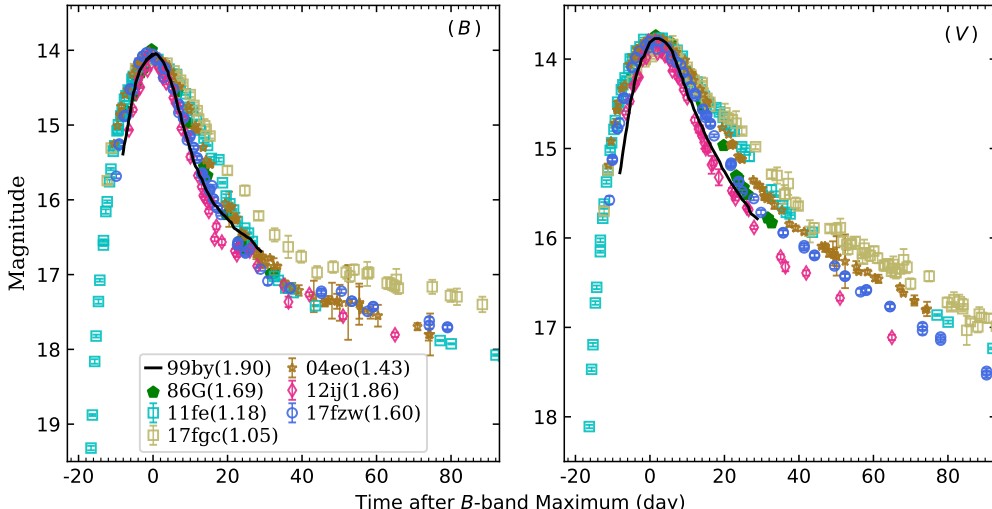

**Figure 4.** *BV*-band light curves of SN 2017fzw from LCO, compared with SNe 1999by (91bg-like subgroup), 2011fe, 2017fgc (normal subgroup), 2004eo, 1986G, and 2012ij (transitional subgroup). The values in brackets represent the light curve shape parameter ($\Delta m_{15}(B)$) [27,28]. All light curves of comparison SNe Ia have been shifted to match SN 2017fzwin magnitudes of the maximum.

In Figure 5, we compare the SN 2017fzw $B - V$ color curve with those of normal and 91bg-like SNe Ia. One can see that the color curve of SN 2017fzw is very similar to that of SN 2012ij in morphology, both are systematically much redder than those of normal SNe Ia. In particularly, SN 2017fzw tends to be even redder than all comparison SNe Ia after the reddest color. After $\sim$+50 days from the *B*-band maximum, all color curves became indistinguishable. On the other hand, the observed $B - V$ color of SN 2017fzw is significantly bluer than 91bg-like SNe Ia (i.e., 0.4–0.7 mag [20]) at the *B*-band maximum light. It is also observed that SN 2017fzw reached the red peak earlier than normal SNe Ia.

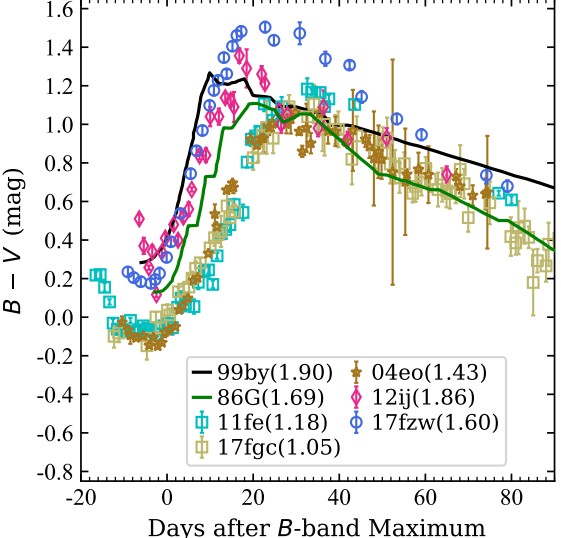

**Figure 5.** $B - V$ color curve of SN 2017fzw, and those of normal SNe 2011fe and 2017fgc, and transitional SNe 2004eo and 2012ij. All SNe Ia have been dereddened for the Milky Way and host galaxies.

The color stretch factor ($S_{BV}$) defined as $S_{BV} = t_{max}^{B-V}/30$ days [56], where $t_{max}^{B-V}$ presents thetime of reddest peak magnitude in the $B - V$ color curve, is proportional to $t_{max}^{B-V}$. Burns et al. (2014) [56] presented a relation between $S_{BV}$ and $\Delta m_{15}(B)$ to estimate the factor $S_{BV}$, but found that the factor $t_{max}^{B-V}$ was better than $\Delta m_{15}(B)$ in the case of fast-evolving 91bg-like SNe Ia, due to the $B$-band light curve of fast-declining SN Ia starts to flatten earlier than about +15 days [56]. Thus, we utilize the former relation to calculate the $S_{BV}$ of SN 2017fzw in this work, and obtain its $S_{BV} = 0.63$ with $t_{max}^{B-V} \approx 19$ days. The estimated $S_{BV}$ is close to some transitional SNe Ia (e.g., $\sim$0.65 for SN 1986G, [23,31]).

### 3.3. Bolometric Light Curves

Based on the response curves of various filters, the quasi-bolometric flux of SN 2017fzw was constructed by trapezoidal integration of flux densities inUV and optical (including $UVW1$, $UVW2$, $U$, $B$, $V$, $g$, $r$, $i$ bands) photometry, as shown in Figure 6. Comparing its quasi-bolometric light curve with those of 91bg-like SNe 1991bg [16] and 1999by [14], transitional SNe 2004eo [61] and 2012ij [25], and normal SNe 2011fe [62] and 2017fgc [63], we find that SN 2017fzw most closely resembles SN 2012ij. Based on the NIR/optical radiative ratios of Type Ia supernova [62,67], the corrections of NIR contribution are taken into account. The modified radiation diffusion model of Arnett implemented in `Minim` Code [68–70] is employed to estimate the radioactive nickel mass of SN 2017fzw as $M_{Ni} = 0.18 \pm 0.03$ M$_\odot$. This is comparable to transitional SNe Ia within reasonable uncertainties (i.e., $\sim$0.14 M$_\odot$, SN 1986G [31] and SN 2012ij [25]). We estimate a maximum luminosity of $L = (4.6 \pm 0.5) \times 10^{42}$ erg s$^{-1}$.

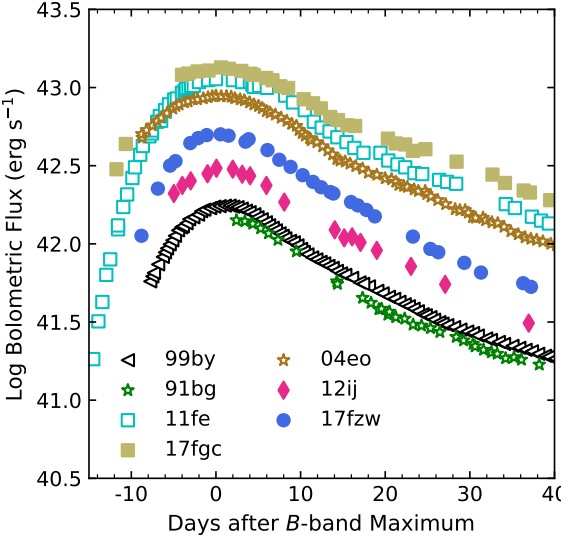

**Figure 6.** Quasi-bolometric (UV and optical) light curve of SN 2017fzw compared with those of normal SNe 2011fe and 2017fgc (Zeng et al. (2021) [63]), and transitional SNe 2004eo and 2012ij (Li et al. (2022) [25]).

## 4. Optical Spectroscopic Properties

Spectroscopic monitoring of SN 2017fzw are displayed in the left panel of Figure 7, spanning the period $-9$ to $+168$ days. The spectral evolution of SN 2017fzw is characterized by several key features found in 91bg-like SNe Ia discussed in the literature, of which we now further elaborate upon [36]. A spectrum of SN 2017fzw at pre-maximum light shows noticeable P-Cygni absorption features of IMEs, such as Si II, S II, Ca II, and Mg II. Its spectra also has a prominent Ti II absorption feature, indicative of 91bg-like SNe, confirming that SN 2017fzw belongs to the subluminous subclass. Furthermore, it has been observed that the Si II $\lambda$5972 and O I absorption features of the spectra of SN 2017fzw exhibit significantly greater strength in comparison to ordinary SNe Ia at corresponding phases. After $\sim$+10 days from the $B$-band maximum, the S II absorption features are

nearly undetectable. In particular, the Si II $\lambda5972$ feature gradually becomes invisible after $\sim+20$ days and is replaced by the Na I feature. However, the Si II $\lambda6355$ feature is still detectable up to about one month past *B*-band maximum light.

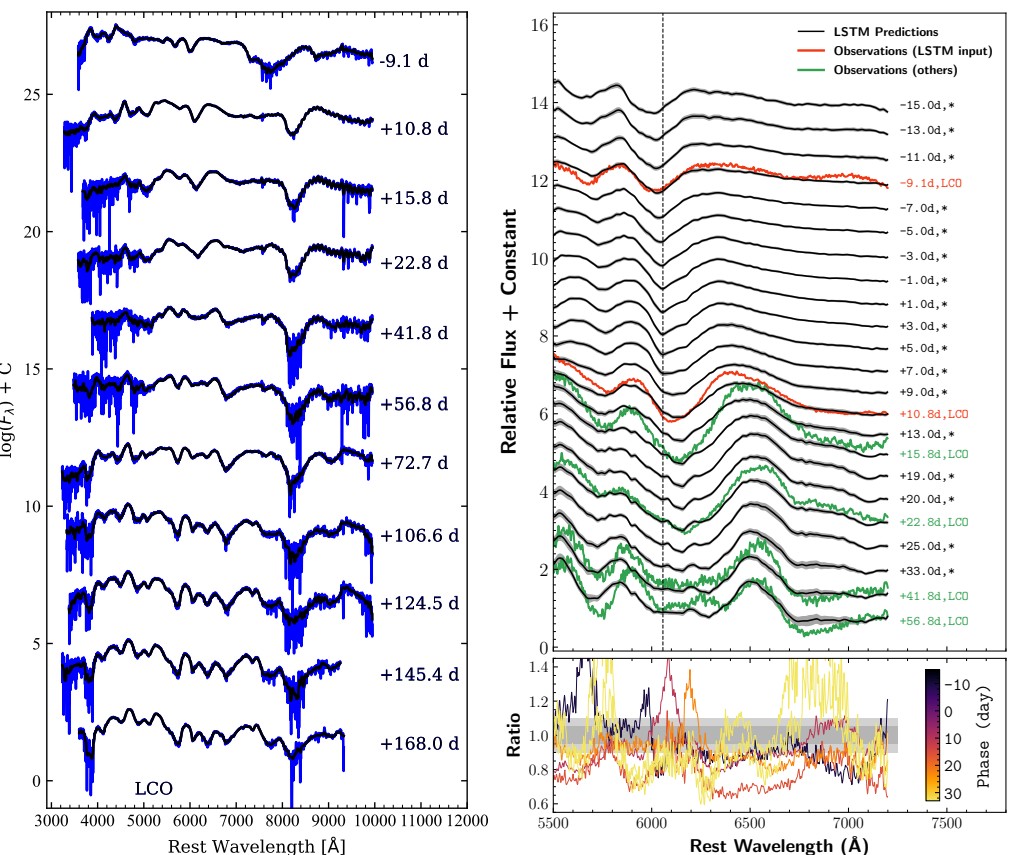

**Figure 7.** Left panel: observed (blue curves) and smoothed (black curves) spectra of SN 2017fzw obtained from LCO. The corresponding phases relative to its *B*-band maximum of this SN are marked on the right side. Right top panel: spectral sequence predicted from two spectra with phase difference ($\Delta p = 20$ days) using long short-term memory (LSTM) neural networks [71] for SN 2017fzw. The predictive mean spectra are marked by solid black curves, the symbol * represents data predicted by deep learning, and their $2\sigma$ standard deviations are indicated by the gray shaded areas. The corresponding observed spectra are plotted as green curves, except for the two input spectra in red. The vertical dashed line indicates the Si II $\lambda6355$ absorption at $-1.0$ day. Right lower panel: flux ratios of predictions to observations for the spectral sequence. All spectra have been vertically shifted for better display.

In Figure 8, detailed spectral comparisons among SN 2017fzw, normal SNe Ia, and transitional SNe Ia are displayed at four phases (i.e., $-10$ d, $+10$ d, $+20$ d, and $+40$ d). The comparison sample include SNe 2004eo [61], 2011fe [62], 2012ij [25], and 2017fgc [63]. The steeper slope of the continuum before the maximum suggests a higher effective temperature of the SN pseudophotosphere than that after the maximum. Like SNe 2004eo and 2012ij, SN 2017fzw shows prominent Ti II and iron-group elements (IGE) absorption features after $\sim+10$ days. It is evident that SN 2017fzw bears a striking resemblance to SN 2012ij with respect to its overall morphology and the intensity of distinctive spectral lines at every phase, with the exception of the Si II $\lambda6355$ and Ca II NIR absorption features. According to Li et al. (2022) [25], the spectra of SN 2012ij resemble those of SN 1999by. Thus, we could say that SN 2017fzw is similar to SN 1999by. These two different features in the spectra of SN 2017fzw tend to resemble SN 2017fgc, which is included in the high-velocity (HV) subgclass due to the classification criteria introduced by Wang et al. (2009) [72]. As shown in Figure 9, at later phases ($\sim+107$ and $+168$ days) SN 2017fzw also tends to develop

characteristics very similar to the HV SN 2017fgc. To summarize, the spectra of SN 2017fzw resembles the 91bg-like SNe Ia at early phases and evolves away from 91bg-like ones at later phases.

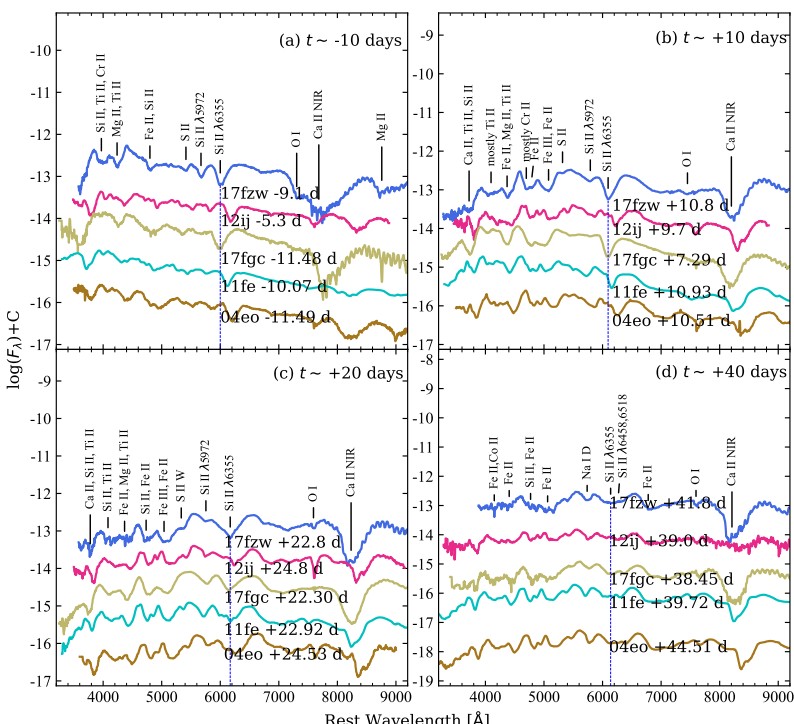

**Figure 8.** Optical spectra of SN 2017fzw at four different phases (i.e., −10 d, +10 d, +20 d, and +40 d) are compared with those of normal SNe 2011fe and 2017fgc, and transitional SNe 2004eo and 2012ij. To enhance clarity, all spectra were vertically adjusted and corrected for both the redshift and reddening of their respective host galaxies, as well as for the Milky Way.

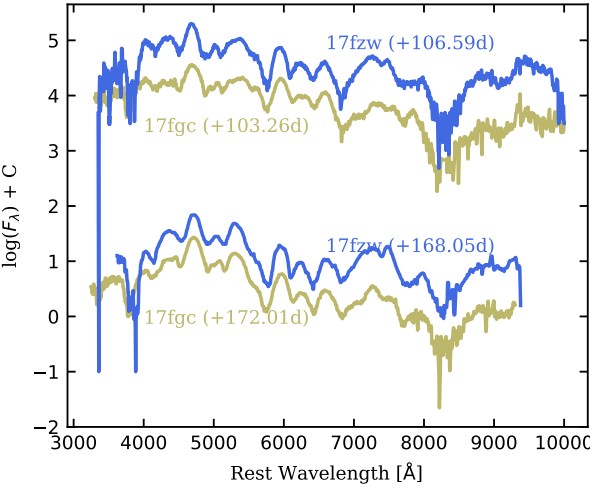

**Figure 9.** Comparison between spectra of SN 2017fzw (blue curve) and SN 2017fgc (yellow curve) at around +105 and +170 days. As in Figure 8, both spectra have been corrected for redshift and reddening.

For SNe Ia, the velocity measured from Si II $\lambda$6355 at *B*-band maximum light is often used to determine their subclass [72]. Unfortunately, SN 2017fzw does not have optical spectral data at around *B*-band maximum light.Thus, the long short-term memory (LSTM) neural networks is employed to predict the spectra of SN 2017fzw in this work [71],

and results are shown in the right panel of Figure 7. One can see that the predictions of SN 2017fzw at around 6355 Å roughly coincide with its observations, except for its spectra after +23 days. For the predicted spectrum of SN 2017fzw at *B*-band maximum, its velocity of Si II $\lambda 6355$ is measured as v = 13,800 ± 415 km s$^{-1}$, significantly beyond the upper limit (i.e., ~11,800 km s$^{-1}$ introduced by Wang et al. (2009) [72]) of the normal-velocity (NV) subclass. We thus classify SN 2017fzw as belonging to the HV subclass like SN 2017fgc. In line with the behavior exhibited by high-velocity Type Ia supernovae, the strength of the Ca II NIR absorption feature in both SN 2017fgc and SN 2017fzw is considerably greater than that observed in the non-high-velocity Type Ia supernovae, namely SN 2004eo, SN 2011fe, and SN 2012ij, after the lapse of one month subsequent to *B*-band maximum light [63].

## 5. Discussion

### 5.1. Transitional Photometric Properties

SN 2017fzw exhibits properties that are characteristic of transitional SNe Ia. Specifically, it has a fast decline rate with $\Delta m_{15}(B) = 1.60$ mag, and an earlier peak time in the *i*-band light curve compared to the *B*-band light curve [32]. Notably, this SN displays a significant shoulder/secondary maximum in *r*/*i*-band light curves, which distinguishes it from 91bg-like SNe Ia. Typically, normal SNe Ia have a stronger shoulder/secondary maximum in NIR light curves, which is associated with the recombination of IGE [73]. Taubenberger (2017) [66] suggested that fast-declining SNe Ia tend to display weaker and earlier shoulder/secondary maximum due to the earlier recombination of IGE. Additionally, in particular types of SNe Ia classified as 91bg-like, it has been observed that the merging of two NIR maxima can lead to the formation of a postponed, solitary NIR maximum. SNe Ia in a transitional phase display characteristics, such as luminosity and temperature, that are intermediary between those of standard and 91bg-like SNe Ia. As such, it is reasonable to suggest that they have a weak shoulder/secondary maximum in their NIR light curves [25].

We estimate the *B*-band absolute peak magnitude ($M_{max}^{B}$) of SN 2017fzw to be approximately $-18.65 \pm 0.13$ mag, which is located at the brightness end of 91bg-like SNe Ia (approximately $-16.5$ to $-17.7$ mag [66]). The $B - V$ color curve of SN 2017fzw is bluer than that of 91bg-like SNe Ia at *B*-band maximum, which is consistent with the findings of Taubenberger et al. (2008) [20]. They discovered that a SN Ia with $\Delta m_{15}(B) \leq 1.75$ mag has a relatively small luminosity, a double maximum in NIR light curves, and a rapid decay.

Figure 10 displays a comparison between $M_{max}^{B}$ ($-18.65 \pm 0.13$mag) of SN 2017fzw and color stretch factor ($S_{BV} = 0.63$, left panel) as well as $\Delta m_{15}$ ($\Delta m_{15} = 1.60$, right panel), utilizing data from Krisciunas et al. (2017) [49] and Li et al. (2022) [25], encompassing overluminous SNe Ia, subluminous SNe Ia, transitional SNe Ia, and normal SNe Ia. It is evident that SN 2017fzw, along with several other SNe categorized as normal or subluminous, is located between the majority of the normal SNe Ia (right-pointing triangles) and the 91bg-like SN Ia (green square). This supports the hypothesis of a continuous distribution of SNe ranging from normal ($S_{BV} \leq \sim 0.5$) to 91bg-like ($S_{BV} \geq \sim 0.8$) [24]. This hypothesis implies that normal and 91bg-like SNe Ia may not originate from two entirely distinct populations.

### 5.2. Transitional Spectroscopic Properties

The spectral evolution of SN 2017fzw is analyzed and compared with normal and transitional SNe Ia in Figures 8 and 9. It is observed that its spectra shift from being more 91bg-like SNe (i.e., having a prominent Ti II absorption feature) at early phases to more normal SNe (mainly HV SN 2017fgc) at later phases, suggesting its transitional characteristics. Notably, SN 2017fzw belongs to the HV normal subclass according to the classification criteria of Wang et al. (2009) [72], which distinguishes it from most transitional SNe Ia (listed in Table 2). This can explain why SN 2017fzw, like HV SN Ia, exhibits a stronger Ca II NIR absorption feature and Si II $\lambda 6355$ velocity compared to the SNe 2004eo and 2012ij.From Table 2, the distribution characteristics of $M_{Ni}$ for transitional SNe Ia are

not obvious due to the small sample in this paper, and a large number of observational studies may be needed in the future to analyze this property.

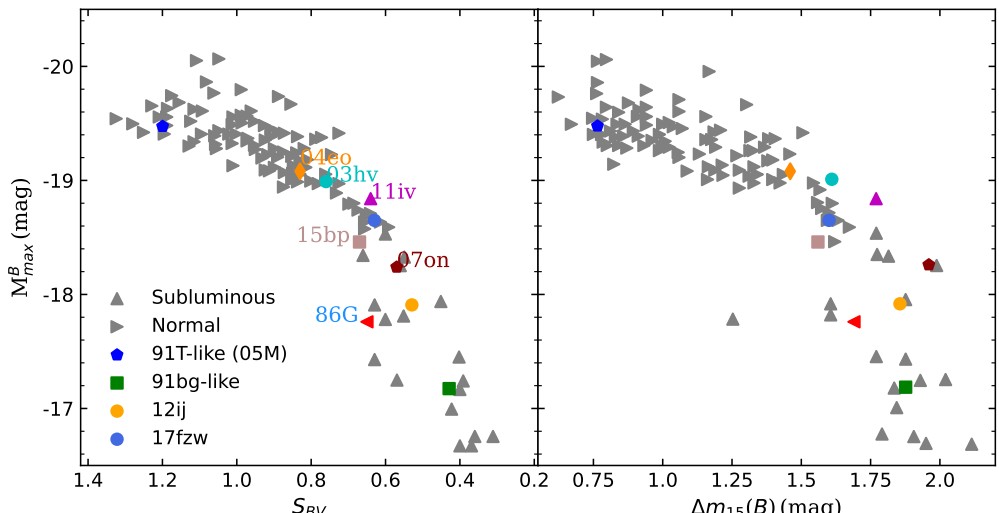

**Figure 10.** Left panel: $M_{max}^B$ of SNe Ia versus their corresponding $S_{BV}$. Right panel: $M_{max}^B$ of the same SNe Ia versus their corresponding $\Delta m_{15}(B)$. The sample includes SN 2017fzw (blue circle), transitional SN 2012ij (orange circle), 91T-like SN Ia (bluepentagon), 91bg-like SN Ia (green square), other subluminous SNe Ia, and normal SNe Ia. The sample is taken from Krisciunas et al. (2017) [49] and Li et al. (2022) [25].

For instance, Li et al. (2022) [25] discovered that transitional SN 2012ij falls into the NV subclass using the same classification criteria established by Wang et al. (2009) [72]. This SN serves as a link between the NV subclass of the normal group and the 91bg-like SNe Ia. On the other hand, SN 2017fzw connects the HV subclass of the normal group with the 91bg-like SNe Ia, which further supports Li et al.'s (2022) [25] finding of a continuous distribution between normal SNe, including the NV and HV normal subclasses, and 91bg-like SNe Ia. Moreover, some authors have also identified a continuous distribution between standard and 91bg-like SNe Ia in other parameter spaces [22,74,75].

**Table 2.** Photometric and spectroscopic properties of transitional SNe Ia.

| Name | $S_{BV}$ | $\Delta m_{15}(B)$ (mag) | $M_{max}^B$ (mag) | $V \times 10^4$ (km s$^{-1}$) | $M_{Ni}$ (M$_\odot$) | SN Type | Reference |
|---|---|---|---|---|---|---|---|
| SN 1986G | 0.65 | 1.69 | $-17.76$ | 0.81 | 0.14 | NV | [23] |
| SN 2003hv | 0.76 | 1.61 | $-18.99$ | - | 0.42 | LVG [1] | [76] |
| SN 2004eo | 0.83 | 1.46 | $-19.08$ | 1.07 | 0.45 | NV | [61] |
| SN 2007on | 0.57 | 1.96 | $-18.24$ | 0.95 | 0.24 | NV | [24] |
| SN 2011iv | 0.63 | 1.77 | $-18.84$ | 0.95 | 0.43 | NV | [24] |
| SN 2012ij | 0.53 | 1.86 | $-17.95$ | 1.05 | 0.14 | NV | [25] |
| SN 2015bp | 0.67 | 1.56 | $-18.46$ | 1.06 | - | NV | [77] |
| SN 2017fzw | 0.63 | 1.60 | $-18.65$ | 1.38 | 0.18 | HV | this paper |

[1] Note that low-velocity gradient (LVG) subclass is proposed by Benetti et al. (2005) [78] base on the velocity gradient of the Si II $\lambda$6355, and most LVG SNe Ia belong to the NV subclass in general [63,72].

### 5.3. Explosion Models

Some observational studies suggest that the violent merger of WD binary may be one explosion model of 91bg-like SNe Ia. During the merging process, the WD's surface is hit by the accretion stream, which leads a high temperature of triggering carbon detonation. Additionally, the lighter companion is disrupted and then the massive WD accretes it in this case [79,80]. Although the total mass of two WDs exceeds the Chandrasekhar limit [81], this case should be considered as a sub-Chandrasekhar-mass (sub-M$_{ch}$) explosion due to

the primary WD's density when WD is ignited [82]. The double detonation in a binary system is another probable explosion channel, which belongs to a sub-$M_{ch}$ explosion as well. In this scenario, the companion star is a helium star or has some helium, and there does not need to a Chandrasekhar mass when WD explodes [83]. These sub-$M_{ch}$ explosions give the results that are suitable for the observations of both narrow, fast-evolving SNe Ia and 91bg-like SNe Ia [84,85].

In addition to the sub-$M_{ch}$ explosions, the $M_{ch}$ delayed-detonation model is also a possible explosion channel. In this case, the initial deflagration at some point will transforms into a supersonic detonation, at which point the nucleosynthesis produced by the explosion is consistent with most SNe Ia, including normal and 91bg-like SNe Ia [85–87].

As proposed by previous works, the all available carbon would be completely burned in sub-$M_{ch}$ explosions, and it is expected that there exists no carbon in the transitional SNe Ia [83,88]. This is in agreement with the undetected carbon of SN 2017fzw at early times. Moreover, according to Wyatt et al. (2021) [77] and Polin et al. (2019) [88], SN 2017fzw in the first group showing a trend between $v$ and $M_{max}^B$, which is related to the sub-$M_{ch}$ explosions. Instead, the $M_{ch}$ explosions is associated with the second group without a notice trend between $v$ and $M_{max}^B$ [89]. Thus, we tend to believe that SN 2017fzw could be originated from sub-$M_{ch}$ explosions.

## 6. Conclusions

In this study, we presented and analyzed the photometric and spectroscopic data of the HV SN Ia 2017fzw with transitional properties. Our analysis revealed that SN 2017fzw has an $M_{max}^B$ of $-18.65$ mag, which is close to the peak magnitude of 91bg-like SNe Ia. Based on its $\Delta m_{15}(B) = 1.60$ mag and color stretch factor ($S_{BV} = 0.63$), we identified SN 2017fzw as a transitional SN that bridges between normal and 91bg-like SNe, thus confirming its transitional nature. Additionally, we classified SN 2017fzw as belonging to the HV subclass with a velocity of $v = 13,800$ km s$^{-1}$. The spectra of SN 2017fzw show similarities to those of 91bg-like SNe, with prominent Ti II features at early phases, and gradually evolve to resemble normal SNe Ia (mainly HV subclass), with stronger Ca II NIR features at later phases.

In conclusion, our analysis shows that SN 2017fzw is a transitional SN Ia with properties that bridge the gap between normal and 91bg-like SNe Ia. This supports the hypothesis that there may be a continuous distribution of SNe spanning from normal to 91bg-like SNe.Furthermore, the progenitor channel of SN 2017fzw tends to be the sub-$M_{ch}$ explosions. The discovery of transitional SN Ia provides an opportunity to investigate the relationships between these two groups, and to gain a better understanding of their progenitors and explosion mechanisms.

**Author Contributions:** Conceptualization, X.Z. and S.Z. (Sheng Zheng); methodology, X.Z. and S.Z. (Sheng Zheng); software, J.H., Y.L., X.Z. and S.Z. (Sheng Zheng); validation, X.Z. and S.Z. (Sheng Zheng); formal analysis, J.H. and Y.L.; investigation, J.H. and X.Z.; resources, S.Z. (Sheng Zheng) and X.Z.; data curation, A.I., K.A.B., D.A.H., C.M. and L.H.; writing—original draft preparation, J.H. and Y.L.; writing—review and editing, J.H., S.Z. (Sheng Zheng) and X.Z.; visualization, S.A.B., A.E., W.L., T.Z., J.Z., S.Z. (Shuguang Zeng), Y.X., Y.H. and L.W. All authors have read and agreed to the published version of the manuscript.

**Funding:** This research was funded by National Natural Science Foundation of China 12203029 and U2031202.

**Institutional Review Board Statement:** Not applicable.

**Informed Consent Statement:** Not applicable.

**Data Availability Statement:** The photometric and spectroscopic data analyzed for this study are downloaded from Las Cumbres Observatory (LCO) and *Neil Gehrels Swift Observatory* (*Swift*).

**Acknowledgments:** The authors thank the staffs of LCO network 1-m/2-m telescopesand *Swift* that observed and provided the data. Furthermore, the LCO group is supported by NSF grants AST-1911225, AST-1911151, and NASAgrant Section 80NSSC19K1639.Additionally, Esamdin is supported by the National Key R&D program of China for Intergovernmental Scientific and Technological Innovation Cooperation Project under No. 2022YFE0126200. Furthermore, this work is supported by The National Natural Science Foundation of China (NSFC, grants 11803076), and the High Level Talent–Heaven Lake Program of Xinjiang Uygur Autonomous Region of China.

**Conflicts of Interest:** The authors declare no conflict of interest.

## Appendix A

**Table A1.** UV and optical photometry of SN 2017fzw obtained from LCO telescopes and SWIFT.

| MJD [1] | *u* | *B* | *V* | *g* | *r* | *i* | *UVW2* | *UVW1* |
|---|---|---|---|---|---|---|---|---|
| 57,974.36 | ⋯ | ⋯ | ⋯ | ⋯ | 17.169(0.009 [2]) | ⋯ | ⋯ | ⋯ |
| 57,978.02 | 16.475(0.054) | 15.784(0.029) | 15.578(0.045) | ⋯ | ⋯ | ⋯ | 18.338(0.090) | 17.890(0.162) |
| 57,981.81 | ⋯ | 15.359(0.025) | 15.124(0.024) | 15.212(0.023) | ⋯ | ⋯ | ⋯ | ⋯ |
| 57,980.17 | ⋯ | 14.982(0.025) | 14.776(0.025) | 14.853(0.021) | 14.611(0.025) | 15.144(0.026) | ⋯ | ⋯ |
| 57,980.76 | 15.599(0.065) | 14.973(0.037) | 14.716(0.055) | ⋯ | ⋯ | ⋯ | 17.910(0.182) | 17.079(0.058) |
| ⋮ | ⋮ | ⋮ | ⋮ | ⋮ | ⋮ | ⋮ | ⋮ | ⋮ |
| 58,156.61 | ⋯ | ⋯ | ⋯ | 18.990(0.045) | ⋯ | ⋯ | ⋯ | ⋯ |
| 58,167.54 | ⋯ | 19.389(0.042) | 19.429(0.057) | 19.178(0.031) | 19.852(0.068) | ⋯ | ⋯ | ⋯ |
| 58,193.51 | ⋯ | ⋯ | ⋯ | 19.725(0.046) | ⋯ | ⋯ | ⋯ | ⋯ |
| 58,215.06 | ⋯ | ⋯ | ⋯ | 20.105(0.041) | ⋯ | ⋯ | ⋯ | ⋯ |
| 58,226.39 | ⋯ | 20.449(0.078) | ⋯ | ⋯ | ⋯ | ⋯ | ⋯ | ⋯ |

[1] Phases are relative to the epoch of *B*-band maximum brightness (MJD = 57,987.90). [2] Uncertainties, in units of 0.001 mag, are $1\sigma$. Note. Measurements are calibrated to the AB magnitude system. (This table is available in its entirety in machine-readable form.)

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
