# Peer review of "SN 2017fzw: A Fast-Expanding Type Ia Supernova with Transitional Features"

_universe, doi:10.3390/universe9060295_

Round 1
Reviewer 1 Report
Dear authors,
I have the following suggestions and questions for a clarification
of your manuscript:
Figure 3:
- right: The big ticks of the x2 and x1 axes of the bottom and
top panels, respectively, are not consistent. Please, correct.
- caption: vars --> bars;
Please, explain the meaning of the legend: Ux3.8, Bx1.5 ...
in the top right panel.
Figure 4:
- What do the numbers in brackets mean? Please, explain in
the caption of the figure.
- Please, specify the shifting - only in the magnitude of
the maximum or also in the time of the maximum.
line 122:
...estimate --> ...estimates
lines 136-139:
This part is not clearly written.
- Does the t^B-V_max time represent the time after the B-band light
maximum? To what it should be proportional?
- "...the factor t^B-V_max was better than Delta(m15,B)..."
This comparison is not clear to me. In what sense should this
comparison be better? Please clarify.
line 153:
If you integrate (reddening-free) fluxes only within the UV/optical
domain, you get only the lower limit of luminosity, because the
radiation outside this wavelength range is not included. From this
point of view, clarify how did you obtain the "maximum" luminosity.
Which distance was used?
Figure 7, the first line of its caption:
- Isn't it the opposite? Observed (blue curve) and smoothed
(black curve)...
A note to Fig. 8:
The steeper slope of the continuum before the maximum suggests
a higher effective temperature of the SN pseudophotosphere than
after the maximum.
Figure 9:
Please correct: ... of SN 2017fzw (yellow curve)... -->
... of SN 2017fzw (blue curve)...
line: 183:
the reference Hu et al. (2022) should be removed (it is given at
the end of the sentence as [68], which is enough).
line 218:
Which overluminous SNe Ia are included in Figure 10?
--------------------------------------
Reviewer 2 Report
I have read the manuscript entitled “SN 2017fzw: A fast-expanding type Ia supernova with transitional features” with great interest. The manuscript discusses the analysis of optical observations of the Type Ia supernova SN 2017fzw. The features of this supernova appear to be consistent with those in the “transitional” category between the 91bg class and normal Type Ia-s.
I find the manuscript informative and well-written. I have two main concerns and several other minor comments.
*******************
Main comments:
*******************
1)
A detailed study of this object already exists in the recent literature. Therefore, in addition to the citation to Graham et al. 2022, it would be beneficial to complement it with a clear discussion of what is known from previous studies. I would recommend dedicating a discussion to the differences between the current and previous studies, including methodologies and main findings. My primary concern is that the results here seem to echo the findings of Graham et al. 2022 without making more explicit statements about their work and comparing the derived parameters. It would be useful if the Authors could clearly highlight the aspects in which this study differs from previous works. It is also worth considering to clarify the language regarding data usage. If the data has been reported and studied before, it should be clearly stated that this work is a re-analysis.
2)
The Authors conclude that “normal” and “91bg-like” SNe may have a common origin (lines 255-256). However, deducing such a statement solely from observing “transitional” objects between the two groups is not straightforward. While the existence of transitional objects is extremely interesting, discussing the physics of the progenitors would be necessary to make such inferences. Therefore, I suggest both i) discussing the differences between the progenitor channels and their observable outcomes, and ii) mitigating the statement about the origin of Type Ia-s.
In particular, some of the following references might be useful additions to the paper:
Clark, et al., 2021, MNRAS, 507, 4367
Barkhudaryan et al., 2019, MNRAS, 490, 718
Maeda & Terada, 2016, IJMPD, 25, 10
Pakmor et al., 2010, Nature, 463, 7277, 61
*********************
Minor comments:
*********************
Table 1: It is unclear which values are obtained in this study and which values are adopted from previous works. Could the Authors please make this distinction? It may also be useful to indicate in this table those values that are found by other authors (e.g., an explicit comparison to Graham et al. 2022, Sand et al. 2019).
General comments on the figures: In some figures, the numbers are missing from the vertical axes. In all instances, I recommend specifying the value of the constant shift. SN 2017zfw is shown with blue color in all figures except Fig. 10. For consistency, it may be good to keep the same color/symbols from figure to figure.
The distance modulus seems to be different than in earlier works. It would be helpful to compare this more directly with previous analysis and discuss it in some more detail since it plays a role in determining the B-band magnitude.
A more elaborate discussion and comparison of nickel masses may be a rewarding avenue. I would suggest including this in Table 2 or adding new a figure that presents this for transitional Type Ia-s.
Reviewer 3 Report
The result of V=13, 786 ± 414 km s−1 looks like a excessive accuracy on background of the error.
Maybe simlpy to round up to : 13,800±415 ?
No special remarks
Round 2
Reviewer 2 Report
Thank you for your work.
Thank you for your work. The newly added text in the manuscript should be carefully revised in terms of language editing before the paper is published.